# Price volatility and GHG emissions analysis on smaller cattle herds typical for the pre-Alpine region, the example of Slovenia

**Jure Brečko**[ID][1]*, **Črtomir Rozman**[2☯], **Jaka Žgajnar**[ID][3☯]

1 Department of Agricultural Economics, Agricultural Institute of Slovenia, Ljubljana, Slovenia, 2 Faculty of Agriculture and Life Science, Department of Agricultural Economics, University of Maribor, Maribor, Slovenia, 3 Biotechnical Faculty, Department of Animal Science, University of Ljubljana, Domžale, Slovenia

☯ These authors contributed equally to this work.
* jure.brecko@kis.si

**Data Availability Statement:** All relevant data are within the manuscript and its Supporting Information files.

## Abstract

Agricultural input and output prices have become extremely volatile in recent years and the global meat industry faces sustainability challenges related to climate change, resource competition, environmental regulations, animal welfare concerns, consumer preferences and industry policies. Additionally, the economic situation of cattle fattening farms has been significantly impacted by two major shocks: the COVID-19 pandemic and the onset of the war in Ukraine. This has led to a growing demand for microsimulation tools that can analyse how these conditions affect the operations of agricultural farms and address various technological challenges at both the farm and sector levels. In this paper, we present a farm model to analyse the cattle farming sector for the pre-Alpine region, using Slovenia, a typical example of this region, as a case study. These farms are particularly important from both social and environmental sustainability perspectives, and it is crucial that economic sustainability follows suit. The results of the SiTFarm model show that, on average, farms in the cattle farming sector achieved modest results between 2018 and 2022, with an average gross margin of 9.57 €/h. However, the variability is significant, with a coefficient of variation 0.74. Only 25% of farms exceeded 17.15 €/h, while 25% did not surpass 4.46 €/h. At the sector level, the gross margin decreased by 12% in 2020 but increased by 99% in 2022 compared to the reference year 2018, highlighting the incredible price volatility over a short period. The model results also indicate greenhouse gas emissions ranging from 5.01 to 7.77 kg $CO_2$ eq. per kg of daily body weight gain on the analysed farms. Nearly half of the farms have GHG emissions for cattle fattening exceeding 6.1 kg $CO_2$ eq. per kg daily body weight gain, while about 10% of farms achieve a sustainability target of approximately 5 kg $CO_2$ eq. per kg of daily body weight gain.

## 1 Introduction

The global economic equilibrium has been significantly reshaped by various economic, medical, political, and financial issues, resulting to pronounced fluctuations in agricultural

**Funding:** The authors received funding by Slovenian Research and Innovation Agency, P4-0133 to JB and P4-0022 to ČR & JŽ.

**Competing interests:** The authors have declared that no competing interests exist.

commodity prices. This volatility has been transmitted throughout the food supply chain, leading to undesirable patterns in marketing margins and disrupting the mechanisms of price transmission [1]. External factors have had a substantial impact on the agricultural sector's economy. The COVID-19 pandemic, in particular, had a negative effect on agricultural systems worldwide. Efforts to control the virus spread, such as restricting movement and interactions, led to economic setbacks, including decreased demand for commercial food services, labor shortages, and reduced handling and production capacity of food and other agricultural products, all of which contributed to a reduction in farmers' output [2].

Food supply security of the agricultural sector remains highly fragile, and the pandemic has introduced numerous threats to its sustainability [3]. Quarantine measures have reduced the availability of labor for critical farming activities, and as the economic crisis triggered by the pandemic deepens, these impacts could become even more severe for agricultural sectors [2]. The volatility of supply chains for agricultural inputs has also intensified, driven by energy crisis and concerns about climate change. The global rise in gas prices and the decline in fertiliser supplies, exacerbated by the war in Ukraine, have further strained energy and agricultural resources within global supply chains. Consequently, the prices of natural gas, oil, and coal have surged rapidly [4]. Simultaneously, the prices of key commodities like grains and vegetable oils have soared to record highs, surpassing even those observed during the global food price crisis over a decade ago. The ongoing war in Ukraine has added further upward pressure on these prices [5].

The global meat industry has also undergone substantial changes due to various factors, including droughts, disease outbreaks, trade policies, investments, and technological advancements. The agricultural industry, particularly the beef sector, faces significant sustainability challenges amid climate change, resource competition, environmental regulations, animal welfare concerns, shifting consumer preferences, and evolving industry policies. The impacts of COVID-19 and ongoing conflicts, such as the war in Ukraine, have further exacerbated these challenges. As a result, increased volatility in cattle numbers, productivity, production, and prices is likely in the coming years. Given the crucial role of product prices in production decisions, analysing price trends and fluctuations is essential. Monitoring global meat prices received by producers is becoming increasingly important [6]. Cattle farms vary in technological adoption, feeding practices, scale of animal husbandry, and strategies to enhance beef production efficiency. These factors are critical prerequisites for sustainable intensification, which is necessary to meet the growing demand for beef [7]. The older EU member states (EU-15) dominate beef production, with their output accounting for 89.9% in 2015, while countries that joined the EU after 2004 (EU-N13, including Slovenia) contributed only 10.1% [8]. Although Slovenia is not a key global beef producer, the sector is extremely important for both social and environmental sustainability.

Slovenia, as an example of a pre-Alpine region, is a geographically diverse, with a significant proportion of the land situated in areas with challenging cultivation conditions. These regions, characterised by natural disadvantages such as difficult climatic conditions, steep slopes, and low soil productivity, are classified as Less Favoured Areas (LFA). LFAs account for about 57% of agricultural land across Europe and are predominantly farmed by extensive cattle and sheep operations [9], with Slovenia having an even higher proportion of around 75% [10]. Agriculture in LFAs is considered highly risky due to these unfavourable conditions [11]. These areas are typically remote and mountainous, exposed to climatic and topographical influences that present additional production challenges compared to intensive lowland farms. Consequently, the performance of farms in LFA is affected by the heterogeneity of the natural and topographical conditions, leading to generally lower efficiency and income compared to lowland farms, with performance varying between farms and across different years [12].

On the other hand, the environmental impact of food production is increasingly significant. Livestock production systems are associated with greenhouse gas emissions (GHG) and have substantially contributed to anthropogenic climate change. Meat consumption is an important source of GHG [13] and beef production, in particular, is known for its high emissions intensity. Beef is often cited as one of the food products with the largest GHG footprint among commonly consumed foods [14]. The proportion of GHG emissions attributable to agriculture varies across EU countries. Between 2005 and 2018, agriculture contributed 9.3% to total GHG emissions in the EU. The highest shares were observed in Ireland (32.7%), Denmark (22.9%) and Latvia (22.3%), while the lowest shares were recorded in Malta (3.0%), Cyprus (5.7%), and Slovakia (6.3%) [15]. In Slovenia, GHG emissions from agricultural activities account for 10.1% of total GHG emissions in 2019, making agriculture the second largest emitting sector after transport. Methane is the dominant contributor, representing 68.4% of total agricultural GHG emissions. This methane is primarily produced during the fermentation of feed in the digestive tracts of domestic animals, especially in the rumen of ruminants, and during the storage of livestock manure [16].

In recent years, there has been an increasing focus on microsimulation models, such as farm models, which facilitate simulations and analyses at the level of agricultural holdings or specific agricultural aggregates [17]. These models are designed to provide valuable insights into the decision-making processes and management practices of individual farms [18]. Additionally, they equip policymakers with a deeper understanding of the dynamics within different types of agricultural holdings, enabling them to make more informed, evidence-based decisions. The rise of result-oriented and data-driven agricultural policies further underscores the importance of micro-simulation tools in assessing the impact of various farm-level policies [18].

Farm models have been developed using different techniques to answer a wide range of questions related to agricultural systems [18]. These models are built on different methodological backgrounds to achieve their specific objectives. The most commonly employed approach is mathematical programming (MP), which includes techniques such as linear programming (LP), nonlinear programming (NLP), mixed integer programming (MIP), and positive mathematical programming (PMP). Additionally, models based on econometric approaches, simulation models, and agent-based models (ABM) have also been developed to enhance the understanding of agricultural systems [19].

Various bio-economic farm models (BEFMs) have been developed to integrate the optimisation of farmers' resource management decisions with quantitative assessments of inputs and outputs. Most of these models rely on the optimisation potential of mathematical programming. BEFMs serve as essential tools for understanding farm-level dynamics and guiding policies that aim to help farms achieve sustainable economic viability, especially during times of price volatility. For instance, the European Commission uses the IFM-CAP model, which is based on the positive mathematical programming (PMP) approach [20]. This model enables the assessment of the impact of various policy measures on specific agricultural aggregates and groups of agricultural holdings across the EU [21]. The IFM-CAP primarily utilizes FADN data and was developed to evaluate the economic and environmental performance of agricultural holdings in response to different measures of the Common Agricultural Policy (CAP). Additionally, other models, such as agent-based models, life cycle analysis, and simulations of environmental impacts in agriculture are also employed for policy assessments strategies. Notably, BEFM models like CAPRI-FT, IFM-CAP, FSSIM, and FARMDYN are frequently used in technology and policy assessments. These models are characterised by the fact that they are used over long periods of time, that they follow the principles of modular design to varying degrees, and that they are generally applicable in a wide range of used cases [22]. As

emphasized by van der Linden et al. [23], despite the often laborious and costly development process, mathematical programming models, once operational, enable relatively quick and inexpensive analysis. These models offer a deeper understanding of decision-making and management at the farm level, providing policy makers with greater insight into individual farm dynamics. This enhanced understanding leads to more informed, evidence-based decisions, leading to greater policy targeting and effectiveness. Numerous reviews of existing agricultural models have demonstrated their usefulness for policy analysis and support. When assessing the economic performance of farms, commonly utilized indicators are revenue, variable costs, and gross margin [24].

The aim of this work is to examine the impact of price volatility (both input and output prices) on economic outcomes at the farm and sector levels during the period 2018–2022, a time characterised by significant upheavals while simultaneously analysing GHG emissions using a farm model. The paper presents (i) a case study that assesses the impact of price volatility on the economic performance of the cattle fattening sector, focusing on typical farms, and (ii) an estimation of GHG emissions from the cattle farming sector in smaller herds typical for the pre-Alpine region, using Slovenia as an example. The methodology employed is briefly outlined, starting with a concise presentation of the farm model, followed by an overview of the typical farms included in the study.

## 2 Material and methods

### 2.1 SiTFarm tool

For our study, SiTFarm (**S**love**n**ian **T**ypical **Farm** Model) tool was used [25]. The modelling background belongs to the field of bioeconomic farm models. The analysis primarily revolves around the farm's production plan, which links the various production activities, taking into account the constraints and assets specific to the analysed farm.

The SiTFarm tool is based on a modular approach. These modules are designed to work independently or to integrate seamlessly into the standardised system as supporting or stand-alone components. It enables various analyses at the level of the production plan of the agricultural holding, groups of farms at the level of individual sectors, as well as the aggregate agricultural sector level. Three modelling approaches are combined in the tool and are presented underneath, namely: i) Model calculations (MC) of the Agricultural Institute of Slovenia; ii) models of typical agricultural holdings (TAH) and iii) Farm model (FM).

**2.1.1 Model calculations–MC.** The SiTFarm tool is fully integrated into a complex system of budget calculations (model calculations) prepared by the Agricultural Institute of Slovenia [26]. The MC, which serve as an independent simulation models for individual agricultural activities, enable the estimation of input usage and, consequently, the calculation of production costs for each agricultural product based on predefined initial technological assumptions. The MCs calculate the consumption of inputs depending on the technology applied, the intensity of production (yield), the size of the plot or herd, the distance of the fields from the farm, the slope of the terrain, and, in some cases, some other technological parameters. The quantity of inputs used are calculated as a function of given production intensity, while the production costs are ultimately calculated as products between the model's estimated inputs usage and their prices. When input quantities are calculated and multiplied with their prices, we obtain a calculation of direct costs and when we add direct and indirect fixed costs, we obtain a model calculation (full-cost enterprise budget). The procedure is called a technological economic simulation model. Actual farm data necessary for cost calculations is rarely available, together with significant limitations of data availability in the planning process, prompting the

utilization of model calculations based on technological economic simulations as a viable solution [27].

This also enables a direct comparison of variable costs with revenues and the calculation of various economic indicators at the level of individual production activities, which together form the farm production plan. Nevertheless, the circumstances in the individual farms may differ significantly from those assumed in the reference MC, the results of which are also published [26]. It is therefore necessary to adapt these as closely as possible to the specific conditions of the individual farms. Consequently, in the present version of the SiTFarm, the budget calculations are adapted to the gross margin (GM) level, which is the focus of the economic analysis in our study. However, in addition to the economic indicators, the MC serves as a simulation tool to determine various technological coefficients ($a_{ij}$) which are linked to the individual production activities in the model and together form the farm's production plan.

*2.1.1.1 Greenhouse gas emissions (GHG) evaluation.* Due to the increased quantities in the atmosphere, greenhouse gas emissions are the main cause of climate change. Dairy cows, beef cattle, and breeding heifers contribute the most to GHG emissions in the cattle fattening sector. GHG emissions from livestock farming include emissions from the rearing of animals on agricultural holdings, while emissions from feed production, including fertilisation with animal fertilisers are not taken into account. The GHG emissions in cattle fattening sector are calculated per kg body weight gain per day. In the calculation and simulation of emissions, several factors are taken into account. The fattening period, daily weight gain, final fattening weight, and the starting weight of the calf for fattening have the greatest influence on the intensity of GHG emissions. Another important factor is the feed ration. It is either based on grass silage and corn silage year-round or animals have also pasture available for a particular period of the year. Another important part of this function is also manure storage technology. It makes a difference whether the manure storage is slurry in lagoons, or technology in stalls is straw bedding.

By increasing breeding efficiency (faster growth, higher daily weight gain), we can significantly reduce the intensity of GHG emissions, i.e. the emissions per unit of processed meat. Optimising feed rations is another effective measure for reducing greenhouse gas emissions.

**2.1.2 Models of Typical Agricultural Holdings–TAH.** There is a growing trend to assess outcomes at the level of typical farms [20]. These farms are usually selected to represent the situation in a specific segment of an agricultural sector. They are reffered to as representative farms and allow generalisations to be made at the aggregate level, providing valuable insights into the overall dynamics of the sector.

TAHs are introduced as the third subsystem of models. The basis for defining the TAH models was the structure of the standard output (SO) of agricultural holdings in 2014 and special workshops conducted as part of another study [28]. These models of farms can be defined as static models that allow the simulation and analysis of various factors at the level of the production plan of an individual farm. The production plan shows the expected situation in a specific type of farm, which is a typical representative of a larger number of agricultural holdings (i.e. it is not a specific agricultural holding, but a typical representative of a specific group of farms). Technically, TAHs are constructed by adjusting the individual MCs.

The level of analysis is the production plan, which forms the basis for analysing conditions at the farm level and later also at the aggregate level (individual sector or the entire agriculture). TAHs are a complex system that lists all conditions at the level of an individual farm. This enables not only the simulation of TAH conditions but also analyses at different levels of the production plan and individual production activities (enterprises) carried out on a given farm.

When determining the TAHs, there are eight types of farming for Slovenia (specialized crop grower, specialized gardener, specialized grower of permanent crops, specialized grazing livestock breeder, specialized pig and poultry farmers, mixed crop production, mixed livestock production, mixed livestock and crop production), which have further subtypes (e.g. for specialized livestock breeders: specialized milk producer, specialized cattle breeder, etc. and in the case of permanent plantations: specialized wine grower, specialized fruit grower, specialized olive grower) similar to the classification in the typology of agricultural holdings at the European Union level.

**2.1.3 Farm model.** In the version of the SiTFarm tool used for the purposes of this study, the tool is based on linear programming (LP). It assumes that the decision-maker (i.e. the farm manager), primarily considers one main objective (maximizing expected gross margin (EGM) at the farm level), taking into account all production constraints along with the characteristics of each TAH.

The primary objective of using LP is not to optimise the entire production plan but to reconstruct the baseline production plan. This reconstruction aims to match the production plan and balance nutrients (ration, fertilizers) and other flows (animal balance, stock balance, produced/sold/purchased fodder) at the farm level with the key information available for each TAH. The resulting solution is the baseline production plan, which is further shocked and analysed with changing exogenous variables (in our case prices in different time frames).

The reconstruction of the production plan is based on defining all key production activities with upper and lower limits. The values of variables ($x_j$) are calculated using the optimisation potential of linear programming (LP), whereby certain activities are given and fixed ($x_f$) and must be included in the optimal solution ($b_f$).

$$\text{maxEGM} = \sum_{j=1}^{n} c_j x_j + \sum_{f=1}^{n} c_f x_f \tag{1}$$

s.t.

$$\sum_{j=1;f=1}^{n;r} a_{ij} x_j \leq b_i \qquad \text{for all } i = 1 \text{ to } m \tag{2}$$

$$x_{f=} b_f \qquad \text{for all } f = 1 \text{ to } r \tag{3}$$

$$x_{j \geq 0} \qquad \text{for all } j \tag{4}$$

All variables that primarily define the farm type, their values are known ($x_f$) (e.g. number of beef) and are fixed $b_f$. These activities remain unchanged in the subsequent analyses. The basic principle is that these activities reflect the production type of the farm. In our example, the coefficients of the objective function ($c_j$ and $c_f$) denote the EGM for each production activity within the model and have been modified with respect to the scenario obtained. Scheme of the SiTFarm model is presented in Fig 1.

## 2.2 Defined typical agricultural holdings (TAH) in the cattle fattening sector

This study presents an analysis of the TAHs that define the Slovenian cattle farming sector. The TAHs were developed in another study [25] and differ in terms of size, production intensity, production orientation and are located in different geographical regions (lowlands, different types of less favoured areas–LFA). There are 12 TAHs with which the Slovenian cattle farming sector could be defined. The key data for each TAH, which is representative of a certain number of real farms is shown in Table 1. The farms are characterised by different degrees

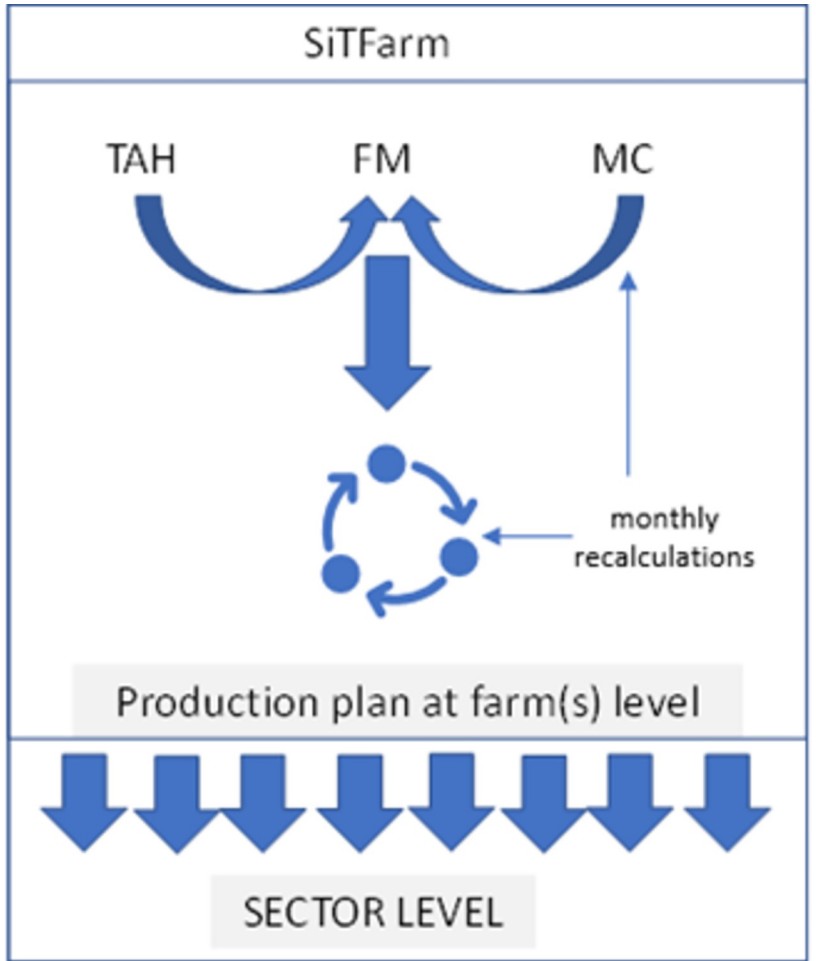

**Fig 1. Scheme of the SiTFarm tool–farm to sector level.**

of specialisation in certain production activities, which primarily correspond to the farming conditions. The range of farms is very wide, and extends from small family farms to larger farms. It is a fairly diverse group in terms of natural resources (field and permanent grassland share), quality and intensity of forage produced, number of animals, and intensity of breeding (daily weight gains vary from 700 up to 1400 g/day). Considering the average size of farms in Slovenia, the cattle fattening sector is typically dominated by relatively small farms and they are dealing with many challenges in meat production.

We present various data for each farm. The main activities on farms, besides cattle farming (bulls for fattening) are fodder production crops (corn, corn for silage, hay, grass silage, pasture, grass-clover mix) and also cash crops such as barley, buckwheat, and hops. For greater informativeness, we divided the name codes of individual TAH into three parts:

- TAH1-12 is the consecutive number of a farm in the cattle sector

- The number in the middle represents the number of beef cattle in the herd on a farm (Table 1)

- The last two letters indicate what type of agricultural land predominates at the farm; ME = meadows, MA = fairly even ratio of meadows and arable land, little more meadows,

**Table 1. The main characteristics of analysed TAHs for the Slovenian cattle farming sector.**

| | | TAH1_1_ME | TAH2_2_ME | TAH3_3_ME | TAH4_6_MA | TAH5_8_AM | TAH6_12_AM |
|---|---|---|---|---|---|---|---|
| *Production activities* | Unit | | | | | | |
| *Bulls for fattening* | heads | 1 | 2 | 3 | 6 | 8 | 12 |
| *Bulls daily weight gain* | (kg/head) | 0.7 | 0.8 | 0.9 | 0.9 | 1.0 | 1.1 |
| *Barley* | (ha) | | | | | | |
| *Corn* | (ha) | | | | 1.13 | 0.56 | 0.93 |
| *Corn for silage* | (ha) | | | | 0.47 | 1.34 | 1.87 |
| | (ha) | | | | | | |
| *Grass clover mixture* | (ha) | | | | 0.40 | 0.48 | 0.7 |
| *Meadows, baled* | (ha) | 0.72 [c] | 1.31 [c] | 1.53 [c] | 1.50 [c] | 0.50 [c] | 0.43 [c] |
| *Meadows, silos* | (ha) | 0.28 [c] | 0.23 [c] | 0.47 [c] | 0.34 [c] | 0.42 [c] | 0.49 [c] |
| *Meadows, hay* | (ha) | | | | | | |
| *Utilized agricultural area on farm* | (ha) | 1.00 | 1.54 | 2.00 | 3.84 | 3.3 | 4.42 |
| *Labor on farm* | (FTE) | 0.1 | 0.2 | 0.2 | 0.2 | 0.2 | 0.2 |
| *Farms* | (No) | 600 | 600 | 600 | 400 | 400 | 450 |
| | | TAH7_17_Am | TAH8_25_Am | TAH9_60_Ma | TAH10_75_Am | TAH11_150_Am | TAH12_150_Am |
| *Production activities* | Unit | | | | | | |
| *Bulls for fattening* | heads | 17 | 25 | 60 | 75 | 150 | 150 |
| *Bulls daily weight gain* | (kg/head) | 1.2 | 1.25 | 1.3 | 1.4 | 1.4 | 1.4 |
| *Barley* | (ha) | | | 2.45 | 1.05 | 2.1 | 2.1 |
| *Corn* | (ha) | 1.67 | 1.96 | 3.68 | 6.51 | 11.15 | 11.15 |
| *Corn for silage* | (ha) | 2.57 | 3.64 | | 9.24 | 20.35 | 20.35 |
| *Buckwheat* | (ha) | | | | 0.21 | 0.42 | 0.42 |
| *Grass clover mixture* | (ha) | 1.06 [d] | 1.40 [b] | | 4.20 [b] | 8.4 [b] | 8.4 [b] |
| *Meadows, baled* | (ha) | 0.33 [c] | 0.69 [d] | 8.91 [d] | 1.32 [d] | 2.04 [d] | 2.04 [d] |
| *Meadows, silos* | (ha) | 0.59 [c] | 0.70 [d] | 0.99 [d] | 0.79 [d] | 1.39 [d] | 1.39 [d] |
| *Meadows, hay* | (ha) | | | | 1.58 [d] | 2.09 [d] | 2.09 [d] |
| *Hops* | (ha) | | | | | | 5.0 |
| *Utilized agricultural area on farm* | (ha) | 6.22 | 8.38 | 16.03 | 24.68 | 47.52 | 52.52 |
| *Labor on farm* | (FTE) | 0.3 | 0.4 | 0.5 | 0.8 | 1.3 | 1.8 |
| *Farms* | (No) | 250 | 250 | 30 | 30 | 18 | 2 |

[a]Three-cut silage-bale

[b]Four-cut silage-silo

[c]Three-cut grass (silage bale, hay bale)

[d]Four-cut grass (silage bale, silo, hay bale).

AM = fairly even ratio of arable land and meadows, little more arable land, Ma = mostly meadows, Am = mostly arable land. Utilization of arable land and grassland within TAHs is presented in Fig 2.

The analysis was conducted by first calculating the average prices for the period 2018–2022 based on monthly input and output prices. On this basis, we calculated baseline production plans for individual farms. Subsequently, we varied the prices with a fixed production plan, calculated on the basis of the cost-price ratios for the period 2018–2022. In this way, we measured the effects of price changes on economic indicators. Thus, for each agricultural holding, 65 results are calculated, which corresponds to the number of monthly periods.

Hectares of arable land and meadows (owned and leased) each farm utilizes are presented in Table 1. Based on the number of animals, size, and structure on farm, there is an estimation

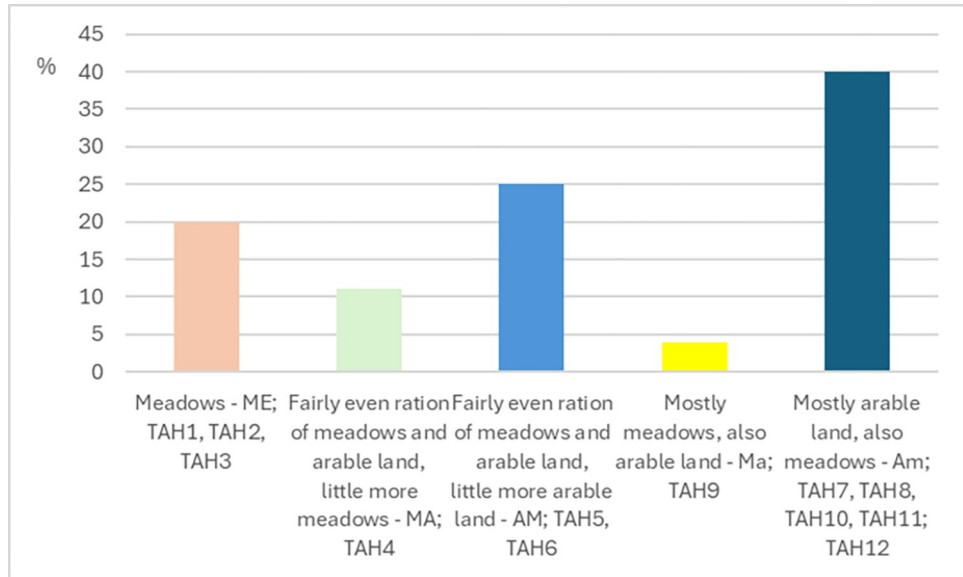

**Fig 2. Utilization of arable land and grassland TAHs are in, in %.**

for the need for effective full-time equivalent (FTE) on each farm. The number of cattle fattening farms represented the Slovenian cattle sector is 3,630. Finally, there is data on how many farms each TAH is representative of (No).

## 3 Results

### 3.1 Farm-level results

We present the main results for the cattle fattening sector in the pre-Alpine region. Firstly, the farm-level results for TAHs are shown and secondly, the economic indicators of the cattle fattening sector are presented. Both results are presented in the context of reaching income sustainability and achieving economic results. Finally, GHG emissions for cattle on TAHs and sector are calculated and presented in the context of the environmental sustainability of farms.

Price data over a five-year period is used for our analysis, with data on a monthly basis for the years 2018–2022. Gross margin per working hour (GM/h) is selected as the main economic indicator at the farm level. It is calculated as the difference between total revenue, which includes subsidies in addition to market revenue, reduced by total variable costs and divided by efficient working hours on a farm.

The TAHs exhibit significant diversity in terms of size, natural characteristics and production intensity, and are mainly focused on breeding cattle and fodder production. However, the economic results achieved have stagnated in recent years and, in some cases, have even declined significantly, with the last two years of our analysis being an exception.

Cattle farming is among the less labor-intensive agricultural activities, which is particularly noticeable on larger farms with efficient organization and investments in fixed assets aimed at minimizing labor input. The majority of farms could operate with labor input of less than 0.5 full-time equivalents (FTE), with only the largest farms exceeding 1 FTE. The range of labor input per animal on farms varies between 0.1 and 0.01 FTE, reflecting different breeding technologies.

Cattle fattening farms utilise only 5% of all arable land in Slovenia. There is a relatively low consumption of phytopharmaceuticals due to less intensive production. An aggregated

analysis of Slovenian agriculture shows that the cattle fattening sector accounts for about around 3% of the total effective labor force in agriculture [25].

COVID-19 restrictions at the beginning of 2020 led to the closures of meat processing plants and a resulting fall in the purchase (output) prices of animals. The closure of state and local borders hindered the mobility and free flow of goods and services, while the procurement of animal fodder encountered considerable difficulties for farmers. A direct consequence of the imposed restrictions were both extremely low purchase prices of live animals and high prices of corn, soy purchased for fodder, and as a result all TAHs achieved the worst economic results in the middle of 2020. The results of TAHs in terms of GM/h are presented in Table 2.

Farms achieve significantly different results due to changes in input and output prices in this timeframe. GM of all farms, calculated as a combined aggregate average in the described period is 9.57 €/h with the smallest (family farm TAH1) achieving only 1.51 €/h GM and the biggest (TAH12) achieving 23.51 €/h. For the larger farms, the cash crops (buckwheat, hops) also significantly contribute to the economic indicators. Farms that were not self-sufficient in fodder production and relied heavily on purchased feed were particularly susceptible to extreme conditions. The minimum calculated on the basis of model calculations was achieved by TAH9 in June 2020 with this timeframe coinciding with the highest feed purchase prices. The coefficient of variation confirms this, as TAH9 having the highest value among all farms at 0.76. We observe significant fluctuations between the minimum and maximum values recorded in specific TAHs, particularly in ones that produce little or no soy and corn feed themselves and must purchase it from the market. These TAHs are extremely vulnerable to price risks, especially in the current volatile evironment.

**3.1.1 Farm-level GHG results.** As expected, each TAH achieve significantly different results in terms of GHG emissions. In Table 3 we see GHG emissions, calculated on a daily and annual basis and as a whole sector for TAH in a smaller market as is pre-Alpine region. Defined standards for emissions in cattle fattening are not fixed but the recommended range is between 5.1 kg $CO_2$ eq. per kg daily body weight gain, and 7.3 kg $CO_2$ eq. per kg daily body weight gain, higher emissions are considered problematic [16]. Although kg $CO_2$ eq. per kg daily body weight gain is an official norm for GHG emissions for cattle, we have also presented the annual emissions per animal and individual TAH. Larger farms achieve better results in terms of kg $CO_2$ eq. per kg of daily body weight gain as they usually have better technologies and balanced feed ration for cattle and follow expert recommendations. The best results in terms of GHG emissions are achieved in TAHs 10, 11, and 12 which use straw bedding manure storage technologies and have higher daily body weight gains which is connected to good feed ration and consequently shorter period for achieving final fattening weight. The worst results are observed in TAHs 1 and 2, which could be characterized as small family farms practicing extensive fattening. These farms typically have calves with lower starting weights, smaller daily body weight gains, and longer fattening periods. Manure is stored as slurry in lagoons, all of which contribute to higher emissions in $CO_2$ equivalent per kilogram of daily body weight gain. It is estimated that the sector as a whole is responsible for 88,354,590 kg of $CO_2$ equivalent per year.

## 3.2 Farm level GM/h changes 2018 to 2022

Prior the Russia-Ukraine conflict, global demand and pandemic-related supply chain disruptions led to a rise in food prices, resulting in a reduction in grain and oilseed reserves and driving price to levels not seen since 2011–2013. Simultaneously, prices for key energy-intensive inputs such as fertilisers remained at record high levels. The invasion exacerbated market disruptions as Russia and Ukraine have historically supplied significant amounts of food imports

**Table 2. Descriptive statistics for GM/h on typical cattle farming TAHs, for the monthly calculated period 2018–2022.**

|  | ALL FARMS | TAH1 | TAH2 | TAH3 | TAH4 | TAH5 | TAH6 | TAH7 | TAH8 | TAH9 | TAH10 | TAH11 | TAH12 |
|---|---|---|---|---|---|---|---|---|---|---|---|---|---|
| Descriptive statistics of GM/h |  |  |  |  |  |  |  |  |  |  |  |  |  |
| Mean | 9.57 | 1.51 | 3.97 | 5.19 | 4.22 | 6.18 | 7.51 | 10.61 | 13.12 | 10.38 | 12.07 | 15.86 | 23.51 |
| Standard Error | 0.25 | 0.04 | 0.11 | 0.14 | 0.27 | 0.27 | 0.33 | 0.38 | 0.48 | 0.99 | 0.65 | 0.81 | 0.47 |
| Median | 7.70 | 1.44 | 3.68 | 4.87 | 3.39 | 5.57 | 6.91 | 9.84 | 12.10 | 7.83 | 10.61 | 14.06 | 22.87 |
| Mode | 4.30 | 1.62 | 4.30 | 5.57 | 5.14 | 6.84 | 8.26 | 11.44 | 14.25 | 12.15 | 12.96 | 16.89 | 23.76 |
| Standard Deviation | 7.03 | 0.31 | 0.88 | 1.09 | 2.19 | 2.16 | 2.64 | 3.03 | 3.85 | 7.88 | 5.19 | 6.48 | 3.77 |
| Coefficient of variation | 0.74 | 0.21 | 0.22 | 0.21 | 0.52 | 0.35 | 0.35 | 0.29 | 0.29 | 0.76 | 0.43 | 0.41 | 0.16 |
| Minimum | 0.68 | 1.08 | 2.96 | 3.87 | 1.40 | 3.27 | 3.78 | 6.40 | 7.99 | 0.68 | 4.85 | 6.82 | 18.17 |
| Maximum | 35.84 | 2.58 | 6.61 | 8.51 | 9.58 | 12.17 | 14.85 | 19.26 | 23.85 | 32.30 | 27.85 | 35.59 | 35.84 |

to developing countries. Since the war began, importing food from the region became more difficult. The ongoing war has further exacerbated the price increase [29]. In this analysis, we have highlighted how global conflicts and price instability impact individual TAHs in a smaller markets, such as those in pre-Alpine regions.

The TAHs representing the economic outcomes of the cattle fattening sector in GM/h are shown in Fig 3 and are separated into two graphs with six TAHs each. The top two graphs present the results for TAHs 1–6 (including relative change and scatter box plot), while the bottom two graphs display the results for TAHs 7–12. The Baseline represents the annual GM/h calculation results for 2018, and we present the relative change in monthly results between 2018 and 2022 in comparison to the Baseline. The graph clearly illustrates significant fluctuations over the years. While the overall trend remains relatively consistent across all farms, individual deviations can be more pronounced on a monthly basis. These relative shifts depend less on farm size and more on factors such as the scale of fodder procurement, the sale of fodder surpluses, and the potential sales of cash crops.

The production plan of the individual farms was not changed during the analysis. Therefore, it is logical that the changes in GM/h show similar movements, but the rate of change we see in the graph is a result of input and output prices in the selected period. Results show extremely volatile agricultural production as GM at the end of 2022 can be up to 300% of those from 2018. At the same time, it is noteworthy to mention how significantly both crises have impacted this sector.

The greatest impact on economic results is seen in the farms that buy the most feed for animals. These types of farms are expected to be the most vulnerable, particularly under conditions such as the two crises that have disrupted agricultural markets. Many farms in the region struggle to provide enough quality feed for the animals due to challenging natural conditions (LFA region). The largest deviation in results are observed in TAHs that need to purchase all

**Table 3. GHG emissions in TAHs.**

|  |  | TAH1 | TAH2 | TAH3 | TAH4 | TAH5 | TAH6 | TAH7 | TAH8 | TAH9 | TAH10 | TAH11 | TAH12 |
|---|---|---|---|---|---|---|---|---|---|---|---|---|---|
| ***GHG emissions on TAH*** | (kg $CO_2$ eq./year) | 4,353 | 8,009 | 10,707 | 17,875 | 22,687 | 29,452 | 43,447 | 61,403 | 169,376 | 196,496 | 392,992 | 392,992 |
|  | (kg $CO_2$ eq./year/ animal) | 4,353 | 4,004 | 3,569 | 2,979 | 2,836 | 2,454 | 2,556 | 2,456 | 2,823 | 2,620 | 2,620 | 2,620 |
|  | (kg $CO_2$ eq./kg daily body weight gain) | 7.77 | 6.90 | 6.15 | 6.77 | 6.17 | 5.58 | 5.22 | 5.54 | 5.15 | 5.01 | 5.14 | 5.14 |
| ***88,354,590*** | (kg $CO_2$ eq./year/sector) |  |  |  |  |  |  |  |  |  |  |  |  |

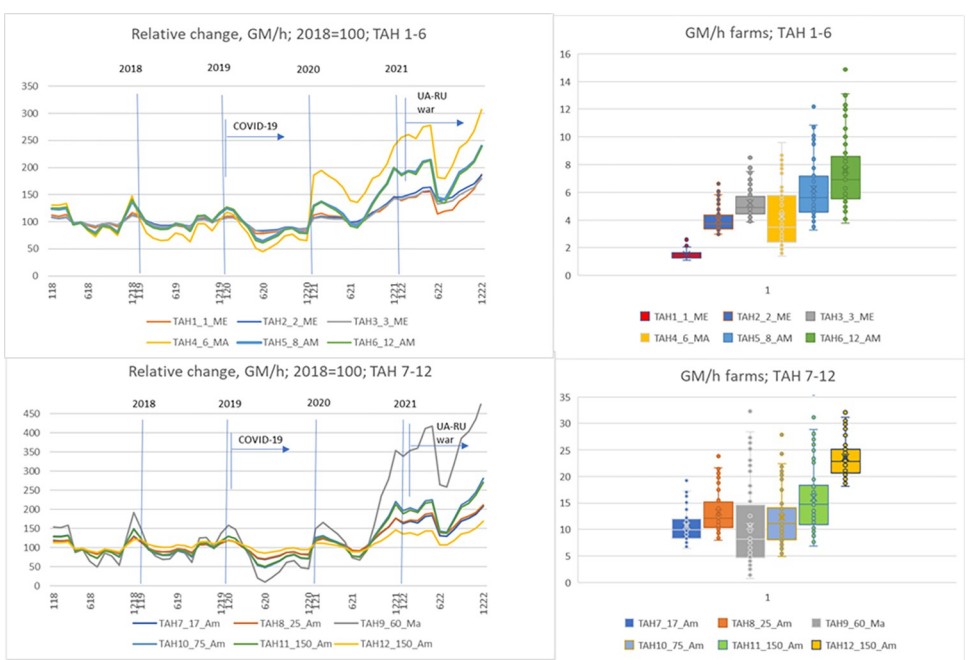

**Fig 3. Relative monthly changes in calculated GM/h on typical cattle TAHs in Slovenia and scatter box plot for GM/h.**

their corn and soy for fodder, such as TAH9. The scatter box plot illustrates the GM/h of these TAHs. Results vary among farms due to different production plans, with no two TAHs having symmetrical whiskers. A wider whisker box plot indicates greater variability within the quartiles. Farms that need to buy most of their fodder, like TAH9, show a larger difference between the minimum and maximum values, reflecting high variability. The scatter box plot analysis reveals that hourly rates generally increase with farm size (ha, number of cattle), and that the differences in hourly rates are significantly greater for larger TAHs.

## 3.3 Sector results

The economic indicators for the cattle fattening sector between 2018 and 2022 are shown in Fig 4 on both a monthly and annual basis. After a relatively stable period in 2018, agricultural input and output prices began to fluctuate. The COVID-19 measures led to negative production impacts and reduced output, causing a sharp decline in prices within the agricultural sector. The calculated GM of the cattle fattening sector was 10.6 million € in January 2018 but fell to 5.8 million € by mid-2020, representing a 45% decrease.

This situation arose due to high fodder prices, as closed borders and COVID-19-related restrictions impacted the availability of fodder, as well as the sales and transportation of live animals. After the COVID-19 restrictions were lifted and the free movement of goods and services resumed, prices for inputs and outputs rose sharply. The volatility of agricultural input supply chains was further exacerbated by energy crisis and fertilizer supply, as a consequence of war in Ukraine early 2022, and the implementation of new green policies addressing climate change. As a result, variable costs, calculated at 43.6 million € in mid-2020, rose to 58.8 million € within two years, an increase of 35%.

The economic indicators for the cattle fattening sector are presented in Table 4, with results shown annually and as an average for the years 2018–2022. Two years show significant

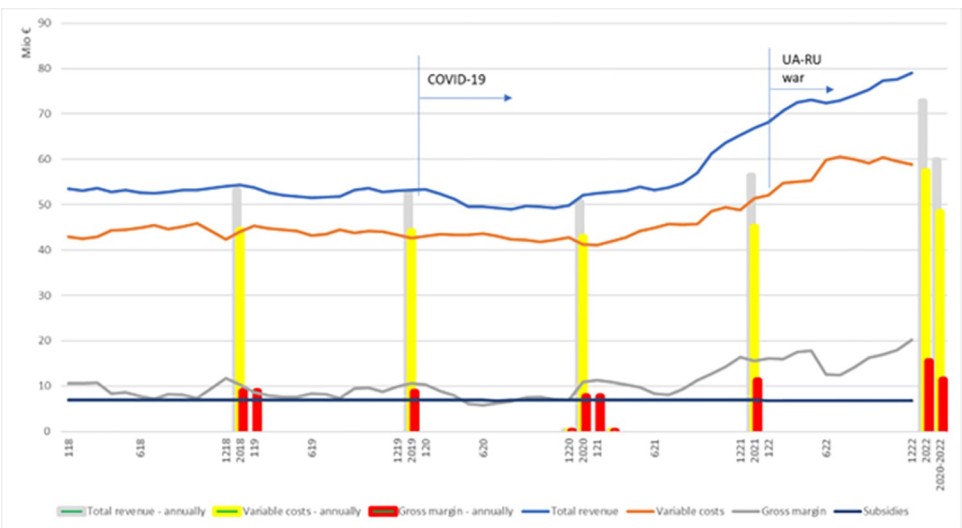

**Fig 4. Economic indicators, including total revenue, variable costs and gross margin for the cattle feeding sector, on a monthly, and yearly basis for the period 2018–2022.**

deviations. First, the 2020 annual results for all economic indicators decline due to the COVID-19 crisis, border closures, and the disruption of the free flow of goods and services. Compared to 2018, revenue decreased by 5%, variable costs by 3.6%, and gross margin by 12%. The second deviation occurred in 2022, which saw substantial increases in all categories. Revenue rose by 44%, variable costs by 34%, and total gross margin nearly doubled, increasing by 99% compared to 2018. This surge is attributed to the lifting of COVID-19 restrictions and the onset of the Ukraine-Russia war, with tariffs and sanctions driving agricultural input and output prices to all-time highs.

## 4 Conclusions

In this study, the SiTFarm model was used to perform a sector analysis on economic indicators for the pre-Alpine region, focusing on the Slovenian cattle fattening sector from 2018 to 2022. The pre-Alpine region has unique characteristics, and 12 typical farms were analyzed, differing in size, production intensity, location and production volume, to represent the cattle fattening sector in the region. These farms are particularly important from both social and environmental sustainability perspectives, and it is crucial that economic sustainability follows suit. None of the TAHs that define this sector could be considered large by EU scale. There are significant differences in the efficiency of TAHs, with the smallest farm achieving only 1.51 €/h GM, while the largest farms, which also cultivate cash crops, achieve up to 23.51 €/h GM.

**Table 4. Economic indicators for cattle fattening sector, annually and as an average for 2018–2022.**

| Year | Total revenue (€) | Variable costs (€) | Gross margin (€) | Gross margin per hour FTE (€/h) |
|------|-------------------|--------------------|------------------|----------------------------------|
| 2018 | 53,202,780 | 44,412,161 | 8,790,619 | 6.14 |
| 2019 | 52,768,883 | 44,104,262 | 8,664,620 | 6.15 |
| 2020 | 50,527,827 | 42,783,302 | 7,744,525 | 5.41 |
| 2021 | 56,310,763 | 45,116,796 | 11,193,967 | 7.82 |
| 2022 | 72,713,559 | 57,313,028 | 15,400,531 | 10.75 |
| 2018–2022 | 57,104,762 | 46,745,910 | 10,358,582 | 7.23 |

The employed methodology has proved to be effective, as it allows for simulations at both the farm and cattle fattening sector level. However, it should be noted that the period analysed was relatively short and characterized by unstable political and economic conditions, making long-term forecasts based on our results unviable. The model is well-suited for analysing the cattle sector in the pre-Alpine region during the defined period. Furthermore, the results highlight the challenges faced by the cattle fattening sector and identify farms within the sector that may be more sensitive to price changes. These findings could also be used for CAP planning, particularly in the development of new measures for the next strategic plan. The cattle farming sector is often seen as industry facing financial difficulties, which justifies production coupled support. Our results reveal significant differences between farms, emphasizing that achieving economies of scale is crucial for better economic performance. Therefore, it is essential that agricultural policy not only includes production coupled payments but also fosters conditions that encourage further investments and promote collaboration among breeders.

We have focused on conducting an analysis that examines the impact of input and output price fluctuations on economic indicators at the farm level. Careful examination of prices over shorter periods is critical, as averaging over longer periods can smooth out the curve and obscure important details. This could pose significant challenges for farmers, who need to respond promptly to price fluctuations. The economic analysis of the cattle fattening sector confirms extremely volatile conditions, particularly after the onset of the Ukraine-Russia war. From 2020 to 2022, model calculations show 44% increase in revenue, while total costs have risen by 34%.

SiTFarm model also enables the quantification and simulation of GHG emissions generated at both the farm and sector levels, reflecting the current situation in the area. Larger farms, which have GHG emissions around 5.1 kg $CO_2$ eq. per kg daily body weight gain, are on track to meet long-term targets for cattle fattening. In contrast, smaller, more extensive family farms exhibit higher emissions, which could be improved through better fodder, changes in fattening technology, and improved manure storage practices. This type of analysis provides decision-makers with a deeper understanding of the dynamics within the sector, facilitating the promotion of sustainability through evidence-based decisions.

## Supporting information

**S1 File.**
(ZIP)

## Author Contributions

**Conceptualization:** Jure Brečko, Črtomir Rozman, Jaka Žgajnar.

**Formal analysis:** Jure Brečko, Jaka Žgajnar.

**Methodology:** Jure Brečko, Jaka Žgajnar.

**Software:** Jure Brečko.

**Supervision:** Črtomir Rozman, Jaka Žgajnar.

**Validation:** Jaka Žgajnar.

**Writing – original draft:** Jure Brečko.

**Writing – review & editing:** Črtomir Rozman, Jaka Žgajnar.

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
