## [Decision Letter · Decision Letter 0]

12 Jul 2024

PONE-D-24-15310Price volatility and GHG emissions analysis on smaller cattle herds typical for the pre-Apal region, the example of SloveniaPLOS ONE

Dear Dr. Brečko,

Thank you for submitting your manuscript to PLOS ONE. After careful consideration, we feel that it has merit but does not fully meet PLOS ONE’s publication criteria as it currently stands. Therefore, we invite you to submit a revised version of the manuscript that addresses the points raised during the review process. Kindly have a look at the reviewer's comments below.

We look forward to receiving your revised manuscript.

Kind regards,

Muhammad Nauman Ahmad, PhD

Academic Editor

PLOS ONE

Journal Requirements:

"The authors acknowledge core financing grants (P4-0133 and P4-0022) of the Slovenian 

Research and Innovation Agency."

"The authors acknowledge core financing grants (P4-0133 and P4-0022) of the Slovenian 

Research and Innovation Agency."

"The author(s) received no specific funding for this work"

6. Please amend the manuscript submission data (via Edit Submission) to include author Dr. Jaka Žgajnar

and Dr. Črtomir Rozman.

7. Please ensure that you refer to Figure 1 and 2 in your text as, if accepted, production will need this reference to link the reader to the figure.

Reviewers' comments:

Reviewer's Responses to Questions

**Comments to the Author**

1. Is the manuscript technically sound, and do the data support the conclusions?

Reviewer #1: Yes

2. Has the statistical analysis been performed appropriately and rigorously? 

Reviewer #1: Yes

3. Have the authors made all data underlying the findings in their manuscript fully available?

Reviewer #1: Yes

4. Is the manuscript presented in an intelligible fashion and written in standard English?

Reviewer #1: No

5. Review Comments to the Author

Reviewer #1: The authors attempt to formulate microsimulation tools to analyse how specific conditions affect farm operations and address different technological challenges at farm and technology sector level. This paper presents a farm model for the analysis of the cattle sector in the pre-alpine region in the context of economic results and ecological effects.

The aim of work is to examine the impact of price volatility (input and output prices) on economic results at the farm and sector level in the period 2018-2022. It was characterised by pronounced upheavals while simultaneously analysing GHG emissions using a farm model. The paper presents (i) an analysis of the impact of price volatility on the economic performance of the cattle sector on typical Slovenian farms and (ii) a parallel estimation of the greenhouse gas emissions of the cattle sector on this type of farm (smaller size).

The authors believe that the methodology used has proved successful as it allows simulations to be carried out at both farm and cattle sector level and can be applied to the analysis of the cattle sector in the pre-Alpine region. I would be more cautious in this type of conclusion. The period analysed is relatively short (5 years), the external conditions (economic, political, environmental) very unstable, the case study concerns the behaviour of small farms, which are the most difficult to predict and very diverse in terms of operating conditions.

In principle, I have no objections to the econometric side of the study (data to be analysed and model), but the theoretical introduction and analysis of the literature on the subject (small livestock farms in the mountain regions of Europe) is far too modest. I am also not sure that combining two issues in one model: the economic viability of livestock farms in the mountains and their environmental impact is not too much to ask. There are few specific statements in the conclusions relating to this interaction.

As an aside: I think there is a mistake in the title: I suppose it is about the Pre-Alpine region, not the Pre-Alpal region (I have not come across this type of term in the geographical literature).

6. PLOS authors have the option to publish the peer review history of their article (what does this mean?). If published, this will include your full peer review and any attached files.

Reviewer #1: **Yes: **Krystian Heffner, University of Economics in Katowice

---

## [Author Response · Author response to Decision Letter 0]

20 Aug 2024

All the comments have been addresed in Response to reviewers.dox, highlighted in red

---

## [Editor Report · Decision Letter 1]

5 Sep 2024

Price volatility and GHG emissions analysis on smaller cattle herds typical for the pre-Alpine region, the example of Slovenia

PONE-D-24-15310R1

Dear Dr. Jure,

We’re pleased to inform you that your manuscript has been judged scientifically suitable for publication and will be formally accepted for publication once it meets all outstanding technical requirements.

Kind regards,

Muhammad Nauman Ahmad, PhD

Academic Editor

PLOS ONE
---

## [Editor Report · Acceptance letter]

13 Nov 2024

PONE-D-24-15310R1 

PLOS ONE

Dear Dr. Brečko, 

I'm pleased to inform you that your manuscript has been deemed suitable for publication in PLOS ONE. Congratulations! Your manuscript is now being handed over to our production team.

Kind regards, 

on behalf of

Dr. Muhammad Nauman Ahmad 

Academic Editor

PLOS ONE